# Protein Translation in the Pathogenesis of Parkinson’s Disease

**DOI:** 10.3390/ijms25042393

**Published:** 2024-02-18

**Authors:** Daniyal Ashraf, Mohammed Repon Khan, Ted M. Dawson, Valina L. Dawson

**Affiliations:** 1Neuroregeneration and Stem Cell Programs, Institute for Cell Engineering, Johns Hopkins University School of Medicine, Baltimore, MD 21205, USA; da521@cam.ac.uk (D.A.); mkhan80@jhu.edu (M.R.K.); 2School of Clinical Medicine, University of Cambridge, Cambridge Biomedical Campus, Box 111, Cambridge CB2 0SP, UK; 3Department of Neurology, Johns Hopkins University School of Medicine, Baltimore, MD 21205, USA; 4Diana Helis Henry Medical Research Foundation, New Orleans, LA 70130, USA; 5Solomon H. Snyder Department of Neuroscience, Johns Hopkins University School of Medicine, Baltimore, MD 21205, USA; 6Department of Pharmacology and Molecular Sciences, Johns Hopkins University School of Medicine, Baltimore, MD 21205, USA; 7Department of Physiology, Johns Hopkins University School of Medicine, Baltimore, MD 21205, USA

**Keywords:** protein aggregation, protein misfolding, α-synuclein, LRRK2, EIF4G1, 4E-BP, PINK1, Parkin, mRNA translation

## Abstract

In recent years, research into Parkinson’s disease and similar neurodegenerative disorders has increasingly suggested that these conditions are synonymous with failures in proteostasis. However, the spotlight of this research has remained firmly focused on the tail end of proteostasis, primarily aggregation, misfolding, and degradation, with protein translation being comparatively overlooked. Now, there is an increasing body of evidence supporting a potential role for translation in the pathogenesis of PD, and its dysregulation is already established in other similar neurodegenerative conditions. In this paper, we consider how altered protein translation fits into the broader picture of PD pathogenesis, working hand in hand to compound the stress placed on neurons, until this becomes irrecoverable. We will also consider molecular players of interest, recent evidence that suggests that aggregates may directly influence translation in PD progression, and the implications for the role of protein translation in our development of clinically useful diagnostics and therapeutics.

## 1. Introduction

Neurodegeneration is a chronic dysregulation of physiological cellular processes in neurons, such as dopamine metabolism, mitochondrial physiology, vesicular transport, membrane physiology, gene transcription, protein translation, folding, degradation, and autophagy. Ultimately, this causes an accumulation of subtle cytotoxic effects, eventually culminating in a cellular burden that precipitates in premature neuronal cell death; in Parkinson’s disease (PD), this presents as a loss of dopaminergic neurons in the substantia nigra, precipitating as the classical clinical presentation of movement symptoms in patients, including bradykinesia, rigidity, resting tremor, and postural instability, among others [1,2].

In recent years, protein aggregation and misfolding have become increasingly synonymous with PD and other similar neurodegenerative disorders, including Alzheimer’s disease, Huntington’s disease, and Creutzfeldt–Jakob disease, to name a few [3]. Certainly, these errors in proteostasis are responsible for the protein aggregates that are the histopathological hallmarks of such conditions, such as the aggregated α-synuclein forming the Lewy bodies characteristic of PD [4,5].

Whilst the majority of work has very much focused on the tail end of proteostasis, i.e., protein quality control, degradation, and autophagy, the focus of this review will be the implications of altered protein translation in PD. Even taking a plain view regarding the formation of protein aggregates, it seems evident that either there is altered synthesis or degradation or, more likely, a combination of the two. Evidence of this can be seen in monogenic forms of Parkinson’s, with defects in mRNA translation linked to both autosomal dominant PD [6,7] and autosomal recessive PD [8], where bulk protein synthesis is coupled to neurodegeneration [9].

Mutations in protein translation leading to neurological disorders are well established, including in autism spectrum disorders [10]; Charcot–Marie disease, through mutations in tRNA synthetase [11]; and cerebellar ataxias, through mutations both in eIF2B (the guanine exchange factor for eukaryotic initiation factor 2, which mediates binding of tRNAi-Met) and ataxin-2 (which regulates mRNA translation through interactions with the poly(A)-binding protein and on mediators of RNA regulation) [12,13]. Indeed, ATXN2 mutations are linked with a PD phenotype, supporting the suggestion that dysregulated protein translation can contribute to the development of PD [13]. In Huntington’s disease, another neurodegenerative disease affecting the basal ganglia, recent work suggests that the polyglutamine expansion of huntingtin, definitive of the condition, promotes ribosome stalling rescued through depletion of the huntingtin protein [14]. Elsewhere, other neurodegenerative conditions, such as Fragile X-associated tremor syndrome and prion-mediated neurodegeneration, are similarly linked to abnormalities in protein synthesis or mRNA metabolism [15,16,17].

Research into the role of translation in the pathobiology of PD and related disorders is, however, comparatively lacking. Recent genome-wide association studies have highlighted loci associated with an increased risk for PD [18], with an increasing body of experimental evidence suggesting that these proteins affect the protein translation process, including leucine-rich repeat kinase 2 (LRRK2), eukaryotic translation initiation factor 4 gamma 1 (EIF4G1), eukaryotic translation initiation factor 4E-binding protein (4E-BP), α-synuclein (SNCA), PTEN-induced kinase 1 (PINK1), and Parkin (PARK2), in a complex network (Table 1) [19]. Specifically, several recessive genes causative of early-onset forms of PD, such as PRKN, PINK1, and DJ-1, are reported as influencing protein translation (Table 1) (these will be discussed in greater detail later). Clearly, the involvement of protein translation in neurodegeneration, including in PD, is not an ungrounded hypothesis.

Here, we will outline the evidence supporting the role of translation as a contributor to neurodegeneration and PD pathogenesis, how perturbed translation may overwhelm neurons leading to their death, molecular players of interest, and the outstanding questions and direction needed in the field.

## 2. How Can Altered Proteostasis Lead to Neuronal Death?

Before discussing individual molecular players and their potential role in PD, it is important to consider how altered proteostasis as a process may lead to such significant cell distress as to cause neuronal loss.

One suggestion is that the energy-expensive nature of protein synthesis would mean increased synthesis affects energy and redox homeostasis, particularly in aged and stressed post-mitotic cells, such as dopaminergic neuron cells, where the energy reserve is low [20]. This, coupled with the mitochondrial dysfunction characteristic of neurodegenerative disorders (discussed in greater detail below), is a likely culprit for causing neuron death.

Furthermore, in such neurodegenerative conditions where altered proteostasis is observed, increased protein synthesis may overwhelm an already compromised or saturated degradation system, leading to an accumulation of misfolded, aberrant proteins precipitating as aggregates. For example, the deregulation of pathways related to eIF2 is reported in the peripheral blood mononuclear cells of both sporadic and genetic PD patients, as well as in prion and, potentially, in prion-like disorders [21,22]. Similarly, studies show an increase in eIF2α phosphorylation in PD patients, closely associated with the accumulation and aggregation of α-synuclein, suggestive of endoplasmic reticulum stress due to misfolded proteins and increased protein translation through the unfolded protein response [23,24]; the phosphorylation of eIF2α also causes the inhibition of global translation [25], potentially representing an attempt by affected neurons to restore proteostasis.

Whilst we hypothesize that the contribution of these mutations in PD pathogenesis is partly due to their role in disturbing proteostasis, it is important to remember that the molecular players we discuss below have multifaceted roles, and thus perturbed protein translation may only partly be responsible for the neurodegeneration that occurs. For instance, part of the pathology induced by mutant proteins is mediated, in part, by the proteins they influence, including those important for mitochondrial function (a key and, comparatively, well researched cause of neurodegenerative disease). An example of this is seen with DJ-1, a deglycase and chaperone protein pivotal in the cellular oxidative stress process and mitochondrial quality control. A mutated form in its prokaryotic homolog displays translational defects in E. coli [26], suggesting a potential compounded effect in neurodegeneration, where a dual disturbance of its energy/redox homeostasis with a failure of the neuron to respond and rescue its own mitochondrial/oxidative stress leads to cell death.

Clearly, these molecular players contribute to the pathogenesis of PD in a number of ways, with altered protein translation being only one such prong. Here, we will focus on the role of these PD-linked loci in translation, referring to other aspects of their role only when relevant in the context of protein synthesis. However, we feel it is important to highlight that, in reality, these mutations will impair a number of cellular processes, working hand in hand to compound the stress placed on neurons until this becomes irrecoverable.

## 3. Leucine-Rich Repeat Kinase 2 (LRRK2)

### 3.1. LRRK2 and Its Canonical Roles

Much focus of research into altered protein synthesis and its role in PD has been on LRRK2, the most common genetic determinant of both familial and sporadic PD. LRRK2 is a large, complex protein kinase that plays a crucial role in various cellular processes, including vesicle trafficking, cytoskeletal dynamics, autophagy, and protein translation (Table 1) [27]. Although it is expressed in various tissues [28,29], higher expression levels are found in the brain, and within neurons, it is found to localize to various cellular compartments, including the cytoplasm, mitochondria [30], endoplasmic reticulum [31], and synaptic vesicles [32]. This, combined with its multi-domain structure, is illustrative of its involvement in diverse cellular functions, with alterations of its activity linked to neurodegenerative processes.

Canonically, the protein is also involved in the phosphorylation of protein translation machinery, such as 4EBP1 [33] and s15 [7], as well as interactions with translation initiation [34], elongation factors [34,35], ribosomal proteins [7], and Argonaute proteins [36]. Thus, current evidence suggests that LRRK2 plays a significant role in the regulation of protein translation, with mutations potentially altering this.

### 3.2. LRRK2 in Parkinson’s Disease and Protein Synthesis

Researchers have identified several causative LRRK2 mutations in familial PD, including G2019S, R1441C/G/H, and Y1699C [37,38], with the G2019S mutation in the kinase domain particularly noteworthy due to its high prevalence in both familial and sporadic PD cases. Studies also report that individuals carrying LRRK2 mutations exhibit an increased risk of developing PD with varying penetrance, suggestive of additional genetic or environmental factors contributing to disease pathogenesis [39].

Early studies in Drosophila suggest that LRRK2 acts to increase global protein synthesis [33] through direct phosphorylation of 4E-BP1. Further work by Martin et al., 2014 [7] demonstrated that LRRK2 regulates phosphorylation of the ribosomal protein s15 (a key player in translation initiation), with phosphorylation associated with altered translation rates and neuronal dysfunction (Figure 1) [40]. In transgenic Drosophila, phospho-deficient s15 acted to rescue dopamine neuron degeneration and the consequent age-related locomotor deficits, linking elevated LRRK2 kinase activity, aberrant protein synthesis, and PD-like symptoms in vivo. Similarly, Drosophila studies show that LRRK2 physically and functionally interacts with protein translation machinery [36]. The authors also observed that altered protein synthesis caused by PD-linked LRRK2 mutations was toxic to the Drosophila neurons, further supporting the hypothesis that dysregulated translation is a potential contributor to neurodegeneration in PD. An actionable mechanism for this has been suggested by Kim et al. [6], who found that this altered protein synthesis resulted in the dysregulation of calcium, finding that the mutant induces translatome alterations in human dopamine neurons in multiple genes involved in calcium regulation, contributing to elevated intracellular calcium levels, a major contributor to PD pathogenesis (Figure 1) [6].

It is important to note that, whilst the focus has been on the role of LRRK2 in increasing protein synthesis, more recent work has suggested that a more nuanced approach is required to understand its role in protein translation and PD. For example, LRRK2-G2019S has been implicated in dysregulated protein synthesis, specifically augmenting RNA translation according to post-mortem data from PD patients, where eIF2α and eED2 phosphorylation changes were measured, indicating a repression in protein synthesis [24]. Recently, Desphande et al., 2020 reported that repressive regulation of protein translation is a proximal function of LRRK2 in PD pathophysiology [41]. Whilst their findings in mammalian cells using models of PD indicated LRRK2-mediated repression of translation, this study also found that human LRRK2-G2019S mutation overexpression in Drosophila showed an increase in translation compared to endogenous WT. The authors suggested this was due to poor sequence homology between Drosophila and human ribosomal proteins compared to human and rat ribosomal proteins, which are 99% identical.

However, the use of post-mortem samples in these papers may not account for a temporal dynamic in the pathogenesis of the disease, with translation dynamics changing with different stages of the disease. Monitoring changes in phosphorylation may not necessarily indicate a straightforward change in protein synthesis but rather attempts by neurons towards the final stages of the disease to change regulatory events and compensate against overactive translation and saturated degradation. Perhaps, the contribution of LRRK2 to PD pathogenesis is dual in nature, contributing to a general increase in global protein synthesis [7,33] and translational repression of key regulators of proteostasis, such as those involved in endolysosomal sorting, mRNA processing, and aspects of the translational mechanism [42]. This would suggest that increased protein synthesis and decreased degradation may both be downstream effectors in LRRK2 G2019S and similar mutations.

### 3.3. LRRK2 and Its Interplay with α-Synuclein in Protein Translation

LRRK2’s actions are not in isolation to other markers, however, with suggestions of an interplay with α-synuclein. Studies show that overactive kinase activity of LRRK2 may be indirectly responsible for α-synuclein phosphorylation, triggering its accumulation and aggregation [43,44]. Discriminant function analysis showed higher levels of α-synuclein in the cerebrospinal fluid of asymptomatic LRRK2 mutation carriers, differentiating them from healthy and symptomatic PD controls [45]. Furthermore, mouse primary neurons expressing the G2019S mutant show increased α-synuclein aggregation, with treatment with HG-10-102-01 (an LRRK2 inhibitor) in α-synuclein transgenic mice reducing levels of α-synuclein phosphorylation and aggregation [46]. However, how does this fit in with a general increase in protein synthesis followed by an observed reduction, whilst still impacting α-synuclein?

One suggestion is that pathogenic LRRK2 disproportionately affects α-synuclein, as postulated by Martin et al., 2014 [7]. A potential mechanism of this can be seen when considering Kim et al., 2020’s findings that 5′UTR complex secondary structures in mRNA are translated more efficiently in G2019S LRRK2 neurons, with earlier findings that such a structure was an important positive regulator of SNCA synthesis [47]. Pathogenic LRRK2 also promotes the oligomerization of α-synuclein on the lysosomal surface, impairing its uptake into the lysosome and, consequently, its subsequent degradation. Furthermore, a slowing down of protein translation in the later stages of PD, as suggested by Deshpande et al., 2020 [41], could potentially be a consequence of stalled degradation, explaining how saturated proteostasis could lead to a global reduction in protein synthesis and be conducive to the formation of α-synuclein aggregates (Figure 1).

### 3.4. LRRK2 and Pharmacological Intervention

The involvement of LRRK2 as a significant player in PD pathogenesis has, consequently and appropriately, raised questions regarding its value as a potential therapy in preventing neurodegenerative phenotypes. LRRK2 inhibitors have shown some efficacy in blocking LRRK2 kinase activity and preventing a neurodegenerative phenotype in various models as part of proof-of-concept studies [48]. However, their use in humans has proven difficult due to safety concerns surrounding off-target effects on other kinases, as well as the potential effects of inhibiting an enzyme that plays important roles in such a wide number of cellular processes. For example, mouse studies have shown pathophysiological changes in the kidneys of mice expressing kinase-dead LRRK2 or LRRK2 knockout [49], as well as pathology in the lung tissue of primates [50]. Other issues have arisen regarding potential effects on the immune system since LRRK2 plays an important immunoregulatory, negatively regulating the nuclear factor of the activated T-cell (NFAT) family of transcription factors, with LRRK2 knockout mice displaying elevated nuclear localization of these transcription factors and increased rates of colitis [51].

There is a need for the development of potent, selective, and non-toxic LRRK2 inhibitors for use in PD that can permeate the blood–brain barrier. More recently, a number of selective LRRK2 inhibitors have progressed to clinical trials. For example, DNL201 showed a lowering of phosphorylation of LRRK2 at serine-935 and of one of its direct substrates, Rab10, at threonine-73 [52]. In cynomolgus macaques, chronic deliverance of pharmacologically appropriate doses did not show negative outcomes, with single and multiple doses showing crossing over into the CSF and tolerance in healthy and PD patients, and modification of downstream lysosomal biomarkers in the latter. Similarly, DNL151, another small-molecule LRRK2 inhibitor, entered clinical trials in 2017, showing good tolerance alongside a dose-dependent reduction in LRRK2 activity by up to 80%, with additional phase 1 studies of radiolabeled DNL151 involving the drug’s pharmacokinetics also completed [53]. Phase 2b trials are currently ongoing, although phase 3 trials have been terminated by the sponsor, with suggestions that the study will be re-designed for a shorter timeline (previous projections suggested the initial trial was due to be completed in 2031).

Micro-RNAs may also provide potential therapeutic strategies, with Wang et al., 2022 showing a mechanism of miRNA regulation of LRRK2, which contributes to PD pathogenesis [54]. The abnormal expression of miR-205 is related to the occurrence of neurodegenerative conditions, including modulating LDL-associated receptor protein 1 in the brain, contributing to the pathogenesis of Alzheimer’s. In PD, miR-205 expression was significantly downregulated coupled with enhanced LRRK2 levels in sporadic PD patient brains, and in vitro studies show that miR-205 acts to rescue neurite outgrowth defects in neurons expressing the PD-related LRRK2 R1441G mutant [55]. Wang et al., 2022 have shown the hypermethylation of the miR-205 promoter region in SH-SY5Y PD model cells, with the inhibition of methylation displaying a reduction in LRRK2 expression [54]. This work suggests that modifying methylation regulation could provide an alternative means of targeting LRRK2 for the treatment of PD.

Similarly, an alternative approach to targeting LRRK2 is the use of antisense oligonucleotides (ASOs), which can facilitate the degradation of LRRK2 by binding to its mRNA. In a mouse model, the intra-cerebral administration of LRRK2-ASOs results in a reduced amount of protein, improving grip strength and decreasing dopaminergic neuron loss and fibril production by endogenous synuclein. Phase I safety trials are currently ongoing for BIB094, an LRRK2-ASO.

### 3.5. Final Thoughts on LRRK2

To suggest that LRRK2 mutations contribute to PD pathogenesis purely through effects on a single process, such as protein translation, is not sensible. Mutations in this multi-purpose kinase likely increase disease burden by disturbing a number of processes, with perturbed protein translation contributing to an increasing burden on a neuron, analogous to the straw that breaks the camel’s back. Likely, one perturbed process could be compensated for, but upsetting so many cellular functions over an extended period of time leads to the cytotoxicity characteristic of neurodegenerative disorders, i.e., multiple independent effects on physiological processes converge on the death of dopamine neurons. This also comes with the additional issue of attempting to delineate the effects of these mutations on a single process, presenting a similar issue as discerning the effects of a “dirty” drug.

Furthermore, as mentioned earlier, some LRRK2 mutations and their contribution to PD pathogenesis show an age-related penetrance, although it is incomplete, even at advanced age. This suggests a role for gene–environment interactions in the toxicity of mutant LRRK2. For example, LRRK2 is known to be an IFN-γ target gene [56], suggesting that inflammation, whether in response to infection or trauma (both precipitating factors in PD), could be a critical player for bridging the effects of genetic and environmental factors in PD pathogenesis. However, this age-related penetrance also highlights an issue with research into PD and neurodegenerative disorders. Generally, the individual mutations and contributors we are looking at are overexpression assays on a short-term basis, which are not reflective of the effects in vivo over a human lifetime. Oftentimes, any data from such (relatively) short-term studies are only minimal/slight, but in patients, these slight effects extended over a period of decades take a once finely tuned dopamine neuron to a tangled web of chaotic signaling and eventually cell death.

## 4. PINK1/PARK2 and Other Molecular Players

### 4.1. PINK1/Parkin and Their Role in PD Pathogenesis

PINK1 (PTEN-induced putative kinase 1) and Parkin (encoded by *PARK2*), encoding a mitochondrially targeted kinase and E3 ubiquitin ligase, respectively, have been implicated in Parkinson’s disease, with mutations in these genes causing autosomal recessive parkinsonism (Table 1) [57]. Similar to LRRK2, mutations in these proteins have been linked to mitochondrial dysfunction, contributing to PD pathogenesis and impacting mitochondrial quality control and cellular homeostasis, with part of these effects likely mediated through the dysregulation of the synthesis of respiratory chain proteins at the mitochondrial outer membrane [58].

ATF4, a key transcription factor controlling the integrated stress response (ISR), which can be activated in response to stresses, such as mitochondrial dysfunction, has been shown to be elevated in PINK1 and the Parkin mutant Drosophila [59]. Interestingly, the ISR constitutes eIF2α phosphorylation, which is mediated by four kinases, with this phosphorylation reducing global protein synthesis whilst promoting the translation of ISR-specific mRNAs [60]. Under basal conditions, it is likely that PINK1 suppresses the ISR through the maintenance of healthy mitochondria.

Nuclear-encoded mRNAs for respiratory chain proteins (nRCCs) are repressed in the cytosol and are recruited to the mitochondrial surface in a TOM20-dependent manner (a mitochondrial import receptor subunit) and facilitated through the binding of PINK1 to the nRCC mRNAs, competing with their translational repressors, as well as enhancing binding of activators, such as eIF4G1 [58]. This study found that PINK1 also physically associates with the mRNA 5′ cap structure in an RNA-independent manner, a process impaired by the PD mutation G309D. Part of this depression is also believed to be mediated through Parkin ubiquitination, e.g., that of the mRNA repressor protein hnRNP-Glo. Perhaps, the action of these mutant genes in PD pathogenesis is twofold, initially leading to perturbed mitochondrial function that cannot be rescued through boosting of local translation of electron transport chain proteins. This, too, highlights how changes in protein translation and RNA metabolism do not affect neurons in PD in isolation, but again, these mechanisms potentiate and facilitate mitochondrial dysfunction, contributing to the disturbance of proteostasis and cellular homeostasis.

Similarly, a number of miRNAs have been found to regulate both Parkin and PINK1 expression. For example, miR-103a-30p, miR-29c, miR-146a, miR-181a, and miR-218 all may play a role in the regulation of Parkin expression in PD patients, showing both protective and pathogenic effects [61,62,63]. One example of an actionable regulatory mechanism was provided by Kim et al., 2016, who reported that miR-27b acted to suppress the expression of PINK1 by binding to its 3-UTR [64]. This provides one potential novel therapeutic strategy of targeting non-coding RNAs to upregulate functional PINK1/Parkin to rescue mitochondrial dysfunction underlying PD pathogenesis. However, such a therapy would not ameliorate phenotype in those with loss-of-function mutations in PINK1/Parkin. Instead, Koentojoro et al., 2016 have previously shown that the mitochondrial receptor Nip3-like protein X (Nix) has a protective role in PINK1- and Parkin-related PD patient cell lines and observed preserved mitochondrial function due to the Nix function in an asymptomatic loss-of-function Parkin mutation carrier who did not develop PD into her eighth decade [65,66]. Upregulating the translation of Nix and alternative mediators of mitophagy through targeting regulatory miRNAs, such as miR-137 [67], demonstrates that understanding protein translation and its regulation in disease provides insight not only into pathogenesis but carries the potential for the development of effective novel clinical therapeutics.

### 4.2. Variable Penetrance of eIF4G1 Mutations May Suggest That PD Pathogenesis Only Precipitates in These Cases if Neurons Are Placed under Adequate Stress

Eukaryotic initiation factor 4 gamma 1 acts as the main scaffold protein for the multi-subunit protein complex eIF4F, facilitating recognition of the mRNA cap structure and recruitment of mRNA to the ribosome [68] and acting to regulate the translation initiation of mRNAs that encode mitochondrial, cell survival, and growth genes in response to various stresses (Table 1) [69]. Mutations in the protein have been linked to the development of both familial and sporadic PD [70,71], with variants also reported as being associated with autopsy-confirmed Lewy body dementia [72], believed to impair the ability of cells to rapidly respond to stress. For example, *EIF4G1 p.A502V* perturbs the binding of eIF4G1 to eIF4E, important for cap-binding processes during protein synthesis, such as the recruitment of mRNA to the ribosome [70,73]. Another mutation, *EIF4G1 p.R1205H*, impairs the ability of eIF4G1 to bind to eIF3e and is believed to mediate binding between the mRNA cap-binding complex and the 40s ribosomal subunit [70,74]. The importance of these complexes to PD pathogenesis can be seen in cases where viruses, such as the flu, have been reported to cleave eIF4G, acting to inhibit cap-dependent cellular protein synthesis and being a precipitating factor in PD [75,76].

Importantly, these variants show an incomplete penetrance, with work suggesting that they are neither a strong nor prevalent risk factor for PD [77]. However, this again illustrates a recurring principle, that the pathogenesis of PD is more complex than a single mechanism of action. Studies have shown that in the presence of DNA damage, eIF4G1 acts to selectively promote the translation of mRNAs involved in cell survival and DNA damage response [78]. Furthermore, eIF4G is also believed to mediate communication between the eIF4F complex and the miRNA-containing silencing complex, participating in miRNA-mediated translation repression [79]. It is believed to be important in protein-clearing pathways, such as the ubiquitin–proteasome system and autophagy, mitochondrial maintenance, dopamine neuron differentiation, and apoptosis, which are all suggested as potential mediators of PD [80]. These functional roles support the importance of this protein in allowing stressed cells to adequately respond and attempt to recover. The variable penetrance of such mutations is likely to represent the presence and severity of other stressors, whether they be environmental or genetic in nature, which place a neuron under stress in the first place, with a PD pathology being precipitated by mutant eIF4G1 due to an inability to adapt and respond.

### 4.3. Translation Factors Such as 4E-BPs Regulate the Ability of Cells to Respond to Stress and Proteostasis

It is important to also consider the role of translation factors themselves, which are often regulated by cellular signaling processes, such as those mediated by the target of rapamycin (TOR), which is well established as going awry in PD [81]. One example of such a factor that has become of increasing interest in PD is that of eukaryotic translation initiation factor 4E-binding proteins (4E-BPs) (although it is important to note that these are likely not a common cause nor strong risk factor for PD) (Table 1). These proteins regulate the function of eIF4E, mediating the ability of a cell to rapidly respond to intrinsic/extrinsic stress and regulate translation accordingly, allowing immediate changes in gene expression from existing mRNAs. This is the rate-limiting step that is highly regulated, with eIF4E competing with eIF4G1 to bind on the dorsal surface of eIF4E, thus disrupting the initiation of translation [82]. This, in turn, is tightly regulated through phosphorylation, with hypo-phosphorylated 4E-BP1 binding to eIF4E with high affinity [83], and, conversely, on hyper-phosphorylation, 4E-BP1 dissociates from eIF4E, allowing for 5′-cap-dependent translation to occur [84]. At the heart of this regulation of 4E-BP phosphorylation is the TOR signaling pathway, which is activated in response to a number of stimuli, including the activation of the PI3K/Akt1 pathway, and can then phosphorylate 4E-BP and other factors to promote cap-dependent translation [85].

A direct link between changes in the function of 4E-BP, protein synthesis, and neurodegeneration can be seen in Huntington’s disease, where hyper-phosphorylation, and thus inactivation, of 4E-BP in the striatum led to aberrant de novo protein synthesis [86]. This study suggested through proteomic characterization, that translation specifically affects sets of proteins, with an upregulation of ribosomal and oxidative phosphorylation proteins and a downregulation of proteins related to neuron structure and function, with the former perhaps contributing to the oxidative stress experienced by neurons. In PD models, Tain et al., 2009 found evidence that the overexpression of the translation inhibitor Thor (4E-BP1), the Drosophila ortholog of the mammalian EIF4EBP1, suppressed the pathologic phenotypes of both Pink1 and Parkin mutants, including dopaminergic neuron degeneration in Drosophila [8]. Interestingly, a loss of the Drosophila LRRK2 homolog also led to an activation of 4E-BP, acting to suppress Pinkl and Parkin pathology in these flies, suggesting that the pharmacological stimulation of 4E-BP activity may hold potential for PD therapy, especially in comparison to LRRK2, which has been traditionally difficult to selectively target without side effects.

Furthermore, the activation of 4E-BP in vivo using the TOR inhibitor, rapamycin, also acted to suppress pathology in these mutants. 4E-BP has also been shown to be inhibited by dominant mutations in LRRK2, the most common cause of parkinsonism. Importantly, the rescue of these neurons was shown to be through a reduction in protein translation rather than the inhibition of other TOR signaling mediated effects, such as inhibition of autophagy downregulation, as evidence that the genetic ablation of 4E-BP was enough to abolish any beneficial effects of rapamycin in vivo, but inhibiting Atg5, a key mediator of autophagy, did not reduce the efficacy of rapamycin-mediated protection. Whilst previously, the potential therapeutic effects of rapamycin in neurodegenerative conditions, such as PD, were suggested due to their role in promoting autophagy, studies have shown that these effects may also be mediated by targeting dysfunctional protein translation, specifically through its reduction [87]. This, too, supports the role of increased protein translation as a contributor to aggregate formation in neurodegenerative conditions, rather than simply being an issue of quality control and autophagy.

## 5. α-Synuclein: An Active Propagator of PD Pathogenesis

α-Synuclein is a presynaptic neuronal protein that is responsible for regulating the synaptic vesicle pool, vesicle trafficking, and subsequent neurotransmitter release (Table 1). These initially soluble monomers misfold and form oligomers that gradually accumulate into insoluble mature fibrils, eventually aggregating into large insoluble fibrils that trigger selective and progressive neuronal death. This is the case not only in PD but other α-synuclein-related neurodegenerative disorders, including dementia with Lewy bodies, multiple system atrophy, REM sleep behavior disorders, and pure autonomic failure [88].

Mutations in the *SNCA* gene, which encodes α-synuclein, cause familial forms of PD and are the basis of sporadic PD risk [89]. Indeed, work by Jowaed et al. [90] reported a reduction in DNA-methyltransferase 1 (DNMT1) in PD patient brains, suggesting that reduced silencing of *SNCA* is detected in the brains of patients with sporadic PD and that the level of α-synuclein expression is an important determinant of PD pathogenesis [90,91]. Further work suggested that α-synuclein acts to sequester the enzyme away from its substrates in the nucleus [92]. This suggests a novel mechanism for epigenetic dysregulation in PD, specifically *SNCA* and global hypomethylation.

Rather than a global rise in protein synthesis contributing to the formation of protein aggregates, however, we see that mechanisms to regulate protein expression are still active in patients with PD [93]. This seems incompatible with a general increase in translation and would also not provide an adequate explanation as to why α-synuclein, especially, forms aggregates. The best answer to this issue is that there may be a preferential use of an extended SNCA transcript in response to cytoplasmic dopamine, with this longer script favoring accumulation and subcellular localization, i.e., the translation of specific transcripts contributes to a pathologic sequence of events (Figure 2) [94]. An alternative mechanism may be that key regulatory miRNAs (miR-7 and miR-153), which inhibit translation of the SNCA transcript through binding to its 3′-UTR, are downregulated; expression analysis of the substantia nigra of patients with PD shows a downregulation of miR-7 and miR-153, where the overexpression of these miRNAs reduced endogenous α-synuclein levels [95,96].

Infection, too, is observed as contributing to an altered translation of α-synuclein and contributing to aggregation, with gastrointestinal infections with norovirus leading to the upregulation of α-synuclein in the enteric nervous system, predisposing it to aggregate formation [97]. Similarly, replication of the influenza virus can induce seeds of aggregated α-synuclein in Lund human mesencephalic dopaminergic cells in vitro but not other proteins implicated in proteinopathy, such as TDP-43 or tau [98]. These neuropathological effects may be potentiated through an inflammatory response, suggesting a possible role of environmental factors, such as infection, in affecting protein translation, specifically α-synuclein, in the early stages of PD pathogenesis.

This concept that α-synuclein functions as a contributor to toxicity reliant on protein translation aligns with the idea that α-synuclein serves as a pivotal toxic protein element in the pathological cascade downstream of other PD proteins [99]. Indeed, the disruption of proteostasis is thought to be critical for pathological α-synuclein toxicity in PD. However, rather than simply being a consequence of disrupted proteostasis, more recent work suggests that pathologic α-synuclein may directly interact with downstream targets that perturb proteostasis. Recent work in our lab has found that pathologic α-synuclein in mice interacts with biological processes, including RNA processing and translation initiation, as well as catabolic processes, such as autophagy [100]. We found that pathologic α-synuclein activates the mammalian target of rapamycin complex 1 (mTORC1), leading to enhanced mRNA translation, protein synthesis, and concomitant neurodegeneration in PD, with the genetic and pharmacologic inhibition of mTOR and protein synthesis in α-synuclein transgenic models rescuing dopamine neuron loss, behavioral deficits, and aberrant biochemical signaling, i.e., reduced protein synthesis is neuroprotective in pathologic α-synuclein models. Specifically, pathologic α-synuclein destabilizes the tuberous sclerosis complex, a negative regulator of the mTOR signaling pathway, leading to enhanced mTORC1 pathway activation and enhanced protein synthesis (Figure 2).

Earlier work by Chung et al., 2017 suggests that monomeric non-pathological α-synuclein inhibits protein synthesis [101]; the findings of this study coupled with our own may suggest that part of the pathological transition in PD is that from monomeric non-pathologic α-synuclein, which decreases protein synthesis, to pathologic α-synuclein, which enhances protein synthesis. This conclusion would agree with the work by Garcia-Esparcia et al., 2015, who found altered machinery of protein synthesis in the substantia nigra and cerebral cortex but preserved protein synthesis pathways in the putamen; these differences corroborated the enriched presence of α-synuclein oligomeric species in the substantia nigra and cerebral cortex, but no such oligomers were detected in the putamen [102].

A question that does remain, however, is that of the chicken vs. the egg: do α-synuclein aggregates propagate the pathogenesis of PD through their downstream effects, or do they arise as a by-product of PD pathogenesis? We suggest that it is the latter, with these aggregates then acting to further potentiate the disease burden on neurons and contribute to the progression of PD. As we continue to delineate the mechanism, it seems increasingly evident that α-synuclein is at the center of propagation of PD pathogenesis, whether that be through affecting mitochondria function (through import and association with the inner membrane, in turn affecting the activity of the electron transport chain and increasing the production of reactive oxygen species), cross-interactions with other pathways, such as TOR, or altering protein synthesis machinery.

## 6. Concluding Remarks and Future Directions

In the present review, we have attempted to highlight the importance of protein translation as a mechanism contributing to the pathogenesis of Parkinson’s disease and other neurodegenerative conditions. However, as we have pointed out throughout, protein translation, or arguably any of the other suggested mediators of PD, does not work in isolation, but instead, cross-interactions between a number of affected cellular processes create a burden on neurons, which eventually becomes irrecoverable. In the case of protein translation especially, it seems that it contributes to the development of PD in two primary ways: contributing to mitochondrial dysfunction (with the mitochondrial surface being an active site of protein translation) or preventing neurons from adapting and recovering from cellular stress.

Despite recent advances in the field, significant amounts of work need to be performed. There has been comparatively little work on the role of protein translation in the pathogenesis of PD, and even less so specifically on the role of translation inside the mitochondria themselves, which may help bring together ideas surrounding proteostasis and mitochondrial dysfunction in PD.

Protein translation may also act as the bridge between environmental factors (such as infection and trauma) and predisposing genetic factors, with these accumulating over a lifetime to contribute to the pathogenesis of PD. Indeed, one area that requires significant work is the link between traumatic brain injury and the development of PD. Perhaps, trauma can affect critical regulatory players in protein translation, analogous to trauma upregulating TDP-43, an RNA-binding protein implicated in other neurodegenerative conditions that is believed to act by altering the translation of specific mRNAs [103].

Interestingly, it has become evident that these α-synuclein aggregates are not simply a by-product of disease pathogenesis but themselves propagate the progression and formation of aggregates and disease through interactions with other proteins and cellular pathways, including those involved in protein synthesis. However, continued work is needed to elucidate how α-synuclein can interact with signaling pathways and players, such as mTOR, as well as the other pathways that it may influence.

When considering whether there is a characteristic increase or decrease in protein synthesis in PD, it seems that this may be mutant specific, with the more important aspect being whether neurons are adequately able to respond and adapt to cell stress. Perhaps, there is a temporal dimension to also be accounted for, where in the later stages of disease with a greater α-synuclein aggregate burden, a vicious positive feedback cycle of protein synthesis is created through the effects of these aggregates on cellular pathways and directly on protein translation machinery, pushing the cell into an anabolic state.

There is also a growing need to have a greater mechanistic understanding of the role of PD-implicated proteins in protein translation processes, with the aim of developing pharmacological agents that can target and slow down the progression of the disease. Part of this will entail considering what the best model to study the effects of PD proteins on protein translation is and whether the findings in these models translate to clinical practice.

One aspect of this process thus far neglected is how altered rates of translation may contribute to misfolding and aggregate formation, as there is a need to monitor protein folding in nascent proteins in the presence of diseased forms of PD proteins. Further work is also needed to understand how PD proteins may impact other processes regulating protein translation, such as the miRNA pathway, which may hold key clues linking protein translation, folding, and clearance together. Perhaps, investigating protein translation in the context of PD may lead to the discovery of the thus far elusive clinical biomarker, which would allow for the monitoring of disease progression in patients.

## Figures and Tables

**Figure 1 ijms-25-02393-f001:**
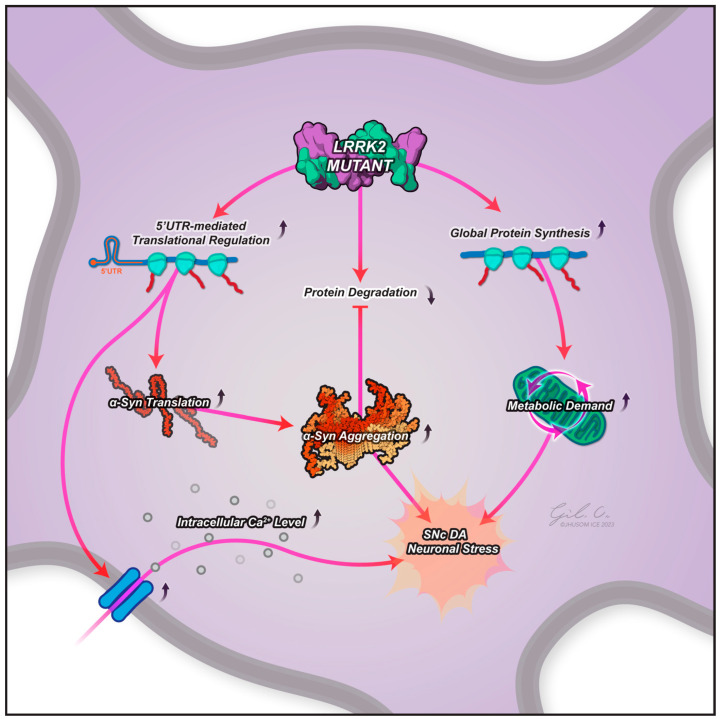
A schematic of the potential contributive effects of an LRRK2 mutant to protein translation in PD pathogenesis. LRRK2 mutants show more efficient translation of mRNA with a complex secondary structure in the 5′-UTR, such as genes involved in calcium regulation (leading to elevated intracellular calcium levels, a known contributor to PD pathogenesis) and perhaps α-synuclein, too. LRRK2 mutants also show a general increase in global protein translation, placing neurons under increased metabolic stress. Evidence suggests that LRRK2 mutants may also lead to decreased protein degradation through translation repression of key regulators of proteostasis (including α-synuclein), as well as being indirectly responsible for α-synuclein phosphorylation, triggering its accumulation and aggregation. These effects may all contribute to the progressive dopaminergic neuronal death observed in the substantia nigra pars compacta in PD. Abbreviations: LRRK2, leucine-rich repeat kinase 2; SNc DA, substantia nigra pars compacta dopamine neuron; 5′-UTR, 5′ untranslated region.

**Figure 2 ijms-25-02393-f002:**
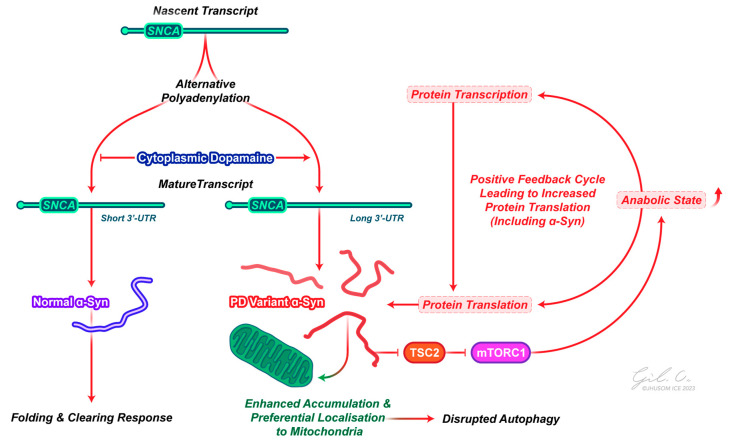
The role of α-synuclein in PD pathogenesis. Nascent α-synuclein transcripts are co-transcriptionally processed to have alternative 3′UTR elements, with elevated cytoplasmic dopamine promoting the generation of an mRNA transcript with a longer 3′UTR. This longer transcript increases the likelihood of the accumulation and localization of α-synuclein protein to the mitochondria, disrupting protein folding and degradation responses and contributing to α-synuclein aggregate formation. In turn, these aggregates can interact with the mTORC1 pathway to promote an anabolic state in the cell, further favoring aggregate build up. Abbreviations: mTORC1, mammalian target of rapamycin complex 1; SNCA, α-synuclein; TSC2, tuberous sclerosis complex 2; PD, Parkinson’s disease; 3′-UTR, 3′ untranslated region.

**Table 1 ijms-25-02393-t001:** A summary of PD risk genes and their suggested contribution to altered protein translation in PD. Abbreviations: EIF4G1, eukaryotic translation initiation factor 4 gamma 1; eIF4E, eukaryotic initiation factor 4E; LRRK2, leucine-rich repeat kinase 2; PD, Parkinson’s disease; PINK1, Pten-induced kinase 1; SNCA, α-synuclein; EIF4EBP1, eukaryotic translation initiation factor 4E-binding protein 1.

Gene	Protein Encoded	Protein Function	Mendelian Inheritance	Proposed Link to Protein Translation in PD
*EIF4EBP1*	Eukaryotic Translation Initiation Factor 4E-Binding Protein 1	Regulates EIF4E function, allowing cells to respond rapidly to stressors through altered translation		Inactivation/loss of function results in aberrant de novo protein synthesis and activation of its Drosophila homolog rescues PINK1/Parkin PD phenotypes
*EIF4G1*	Eukaryotic Translation Initiation Factor 4 Gamma 1	Facilitates recognition of the mRNA cap structure and recruitment of mRNA to the ribosome	Late-Onset, Autosomal Dominant PD	Impairs cap-dependent cellular protein synthesis, impairing the ability of cells to respond to stress
*LRRK2*	Leucine-Rich Repeat Kinase 2	Large protein kinase is involved in a number of processes, including vesicle trafficking, cytoskeletal dynamics, autophagy, and protein translation	Late-Onset, Autosomal Dominant PD	Increase in global protein synthesis with a disproportionate effect on α-synuclein, as well as phosphorylating α-synuclein, leading to its accumulation and aggregation
*PARK2*	PARKIN	E3 ubiquitin ligase	Early-Onset, Autosomal Recessive PD	Dysregulates translation of respiratory chain proteins at the mitochondrial outer membrane
*PINK1*	Pten-Induced Kinase 1	Mitochondrially targeted kinase, important in mitochondrial quality control	Early-Onset, Autosomal Recessive PD	Dysregulates translation of respiratory chain proteins at the mitochondrial outer membrane
*SNCA*	α-Synuclein	Regulates synaptic vesicle trafficking	Autosomal Dominant PD	Alternate transcript utilized in PD pathogenesis and propagated increased rates of protein translation through interaction with downstream molecular players

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
