# Peer review of "Protein Translation in the Pathogenesis of Parkinson’s Disease"

_ijms, 2024, doi:10.3390/ijms25042393_

Round 1

Reviewer 1 Report

Comments and Suggestions for Authors

This review from Ashraf et al nicely summarized recent findings on the potential pathogenic role of altered protein translation in Parkinson’s disease (PD). Molecular pathways that may regulate protein translation by PD-related mutations/proteins such as LRRK2, PINK1, PRKN, eIF4G1, 4E-BP, and alpha-synuclein were particularly described in detail. The author also highlighted that protein translation, or any suggested mediators of PD, do not work in isolation, and need to be studied together to better understand the disease progression. The current review is well written and brought up a relatively under studied but important field.

As mitochondrial dysfunction is one of the primary contributors to PD development, in addition to translation of nuclear protein, perhaps the authors can also touch upon the regulation of protein synthesis/translation inside mitochondria and its potential implication in PD.

Author Response

Thank you for your review. We have added in a small section in the PINK1/PARKIN section about the role of miRNAs in maintaining mitochondrial integrity and protein translation. We didn’t go into excessive detail here, as we felt it would detract from the overall focus on protein translation, and otherwise may get confusing with reference to mitochondrial dysfunction but agree it is important to mention. However, a key issue here was that there isn’t much literature surrounding translation regulation inside the mitochondria, and where there is, it tends to be vague as to general mitochondrial dysfunction: we have highlighted this as an area for further investigation in the conclusion.

Reviewer 2 Report

Comments and Suggestions for Authors

In the article entitled "Protein Translation in the Pathogenesis of Parkinson's Disease," the authors discuss the importance of protein translation in the development of Parkinson's disease (PD) and other neurodegenerative conditions, shedding light on its role in aggravating stress on neurons and potentially contributing to PD progression.

They discuss the potential influence of protein aggregates on translation in PD, suggesting that aggregates may directly impact translation and further contribute to disease pathogenesis.

The article also explores the implications of altered protein translation for the development of clinically useful diagnostics and therapeutics for PD.

Additionally, it emphasizes the need for a greater mechanistic understanding of the role of PD-involved proteins in protein translation processes to develop pharmacological agents that can slow disease progression.

The paper raises the question of the best model to study the effects of PD proteins on protein translation and whether the findings in these models translate into clinical practice.

It is an interesting and important review on the topic of Parkinson's disease.

It is recommended that in each section where they describe the proteins involved in this process such as LRRK2, PINK1/PARK2, alpha-synuclein, they include non-coding RNAs in the regulatory pathway.

 This would give a better understanding of prosteostasis in PD.

Author Response

Thank you for your review. We have incorporated non-coding RNAs, especially miRNAs, into LRRK2 (pharmacological interventions), PINK1/PARKIN and alpha-synuclein.

Reviewer 3 Report

Comments and Suggestions for Authors

This paper is an extraordinary review on protein translation in PD! The authors did a great job in figure creation and organizing the information in an organized way. 

One of the only suggestions I would make is perhaps make a table with all the PD risk genes with their mutations and indicate which mutation could or does contribute to blocking a-syn degradation. This way, the reader and look quickly at what genes and mutations are important in understanding proteostasis. 

As more of a question than a suggestion, would any viruses/bacteria or other causes of brain trauma cause problems as well in a-syn degradation? I know LRRK2 is responsive to IFNg, so perhaps any inflammation occurring can exacerbate problems with a-syn degradation? If this is a possibility, maybe include a small section on environmental problems alongside these mutations could cause a-syn build-up? This way, connection between genetic and environment could be made for pathogenesis of PD.   

Overall, this is a well written paper and has great detailed on the information given. Pretty much is ready to go for publication after looking/adding in my suggestions! 

Author Response

Thank you so much for your review!

We have added in a table where we summarise the key genes we discuss and the mechanisms through which they may contribute to altered protein translation in PD. We have placed this at the end of our introduction.

Concerning the intersection between environment and genetics, we have attempted to make reference to this throughout. We have added in a few comments about LRRK2 and its potential role in inflammation in response to trauma/infection in section 2.5. Similarly, we also mentioned the role of infection in altering alpha-synuclein translation in PD. There isn’t a lot of work on this area, but we felt it was important to highlight that protein translation may be what’s missing in our view of how environment and genetics interact in PD pathogenesis, and discuss this in our conclusion.

Round 2

Reviewer 2 Report

Comments and Suggestions for Authors

Thanks to the authors for the replies. The article improved substantially with the changes done.